# Fractalkine Improves the Expression of Endometrium Receptivity-Related Genes and Proteins at Desferrioxamine-Induced Iron Deficiency in HEC-1A Cells

**DOI:** 10.3390/ijms24097924

**Published:** 2023-04-27

**Authors:** Edina Pandur, Ramóna Pap, Gergely Jánosa, Adrienn Horváth, Katalin Sipos

**Affiliations:** 1Department of Pharmaceutical Biology, Faculty of Pharmacy, University of Pécs, H-7624 Pécs, Hungary; pap.ramona@pte.hu (R.P.); janosa.gergely@gytk.pte.hu (G.J.); horvath.adrienn2@pte.hu (A.H.); katalin.sipos@aok.pte.hu (K.S.); 2National Laboratory on Human Reproduction, University of Pécs, H-7624 Pécs, Hungary

**Keywords:** fractalkine, endometrium receptivity, iron deficiency, cytokines, signaling

## Abstract

Fractalkine (CX3CL1/FKN) is a unique chemokine belonging to the CX3C chemokine subclass. FKN exists in two forms: a membrane-bound form expressed by both endometrium cells and trophoblasts thought to be implicated in maternal–fetal interaction and a soluble form expressed by endometrium cells. Endometrium receptivity is crucial in embryo implantation and a complex process regulated by large numbers of proteins, e.g., cytokines, progesterone receptor (PR), SOX-17, prostaglandin receptors (PTGER2), and tissue inhibitors of metalloproteinases (TIMPs). It has also been reported that iron is important in fertility and affects the iron status of the mother. Therefore, iron availability in the embryo contributes to fertilization and pregnancy. In this study, we focused on the effect of iron deficiency on the secreted cytokines (IL-6, IL-1β, leukocyte inhibitory factor, TGF-β), chemokines (IL-8, FKN), and other regulatory proteins (bone morphogenic protein 2, activin, follistatin, PR, SOX-17, prostaglandin E2 receptor, TIMP2), and the modifying effect of FKN on the expression of these proteins, which may improve endometrium receptivity. Endometrial iron deficiency was mediated by desferrioxamine (DFO) treatment of HEC-1A cells. FKN was added to the cells 24 h and 48 h after DFO with or without serum for modelling the possible iron dependence of the alterations. Our findings support the hypothesis that FKN ameliorates the effects of anemia on the receptivity-related genes and proteins in HEC-1A cells by increasing the secretion of the receptivity-related cytokines via the fractalkine receptor (CX3CR1). FKN may contribute to cell proliferation and differentiation by regulating activin, follistatin, and BMP2 expressions, and to implantation by altering the protein levels of PR, SOX-17, PTGER2, and TIMP2. FKN mitigates the negative effect of iron deficiency on the receptivity-related genes and proteins of HEC-1A endometrium cells, suggesting its important role in the regulation of endometrium receptivity.

## 1. Introduction

Endometrium receptivity is crucial in embryo implantation [1,2]. The preparation of the endometrial cells for embryo attachment is a difficult and complex process influenced by several different factors and circumstances [2,3]. Two thirds of the implantation failure of the blastocysts is associated with decreased endometrial receptivity [4]. The treatment options for impaired endometrial receptivity are limited. Therefore, further research is necessary for finding possible molecular targets to participate in the receptive function of the endometrium [5,6].

Iron is an essential element for all living organisms, and it is also implicated in fertilization [7]. For embryo development, iron availability from the mother is a necessity. Thus, the maternal iron status is fundamental [7,8,9]. Iron deficiency affects cell division and pattern formation, as well as the development and maturation of the central nervous system, since iron is crucial in myelin sheath formation around the neuronal axons [9,10,11,12]. Maternal iron disorders increase retardation of embryonic development and/or lethality due to faulty iron transport through the placenta [12,13]. Iron deficiency anemia affects pregnancy by decreasing the growth and development of the fetus [14].

Endometrial cells secrete a large number of cytokines and regulatory factors for growth, proliferation, and differentiation [15]. These molecules can act not only on the endometrium cells in an autocrine way, preparing them for implantation, but also on the trophoblast cells surrounding the embryo [2]. Interleukin-1β (IL-1β) may act in a synergistic way with estradiol and progesterone to improve receptivity [16]. Leukocyte inhibitory factor (LIF) is an interleukin-6 (IL-6) family cytokine regulating intracellular signaling pathways e.g., mitogen-activated protein kinase (MAPK), Janus kinase (Jak)/STAT or phosphatidylinositol 3-kinase (PI3K) via its receptor, LIFR [17]. Endometrial LIF expression is elevated around the implantation, suggesting its role in receptivity [15]. LIF expression is regulated by transforming growth factor-β (TGF-β) and activin A, as well as by IL-6, which are also involved in the development of endometrium receptivity [15,18,19]. LIF can induce the secretion of prostaglandins of trophoblast cells having an important role in successful implantation [20,21]. On the other hand, the prostaglandin E2 receptor (PTGE2R) expressed by the endometrium is activated by prostaglandins and mediates the development of endometrium receptivity [22]. Moreover, prostaglandins can modify the expression of tissue inhibitors of metalloproteinases (TIMPs) to regulate the invasion of the embryo by controlling the activity of matrix metalloproteinases (MMPs) [23,24]. TNFα pro-inflammatory cytokine secreted by the endometrium protects uterine tissue against increased trophoblast invasion and regulates the synthesis of MMPs [25,26]. The elevated level of secreted IL-8 from the endometrium can stimulate trophoblast migration and invasion for embryo implantation [27,28].

The role of chemokines in the development of endometrial receptivity, embryo implantation, and pregnancy is not fully understood [29,30].

Fractalkine (FKN) is a special chemokine that exists in two forms: a membrane-bound form expressed by both endometrium cells and trophoblasts thought to be implicated in maternal–fetal interaction and a soluble form expressed by endometrium cells [31,32]. In our previous studies, we revealed that the fractalkine (CX3CL1)/fractalkine receptor (CX3CR1) axis regulates endometrium receptivity as well as the iron metabolism of the HEC-1A endometrium cells and triggers iron transport towards the embryonic trophoblast cells in an in vitro co-culture model [33,34].

The FKN/CX3R1 interaction controls different signaling pathways, such as MAPK, protein kinase C (PKC) and nuclear factor kappa-light-chain-enhancer of activated B cells (NFκB) [35]. The FKN-mediated NFκB pathway modifies the expression of cytokines related to endometrium receptivity [36,37]. The NFκB signaling pathway can be altered by the Nrf2/Keap-1 transcription factor system, which is regulated by the progesterone receptor (PR) [34,38,39]. The CX3CR1 activation also increases the transcription of SOX-17 via PR, regulating receptivity and embryo implantation [40].

The role of FKN in pregnancy is controversial. FKN level is elevated in the first trimester in the first trimester of pregnancy, but at high concentrations, it may associate with preeclampsia [41], although the serum level of FKN cannot be used as a prediction marker for preeclampsia [42]. Maternal serum FKN level may contribute to pregnancy complications like fetal growth restriction [43]. On the other hand, decreased expression of FKN in granulosa cells may contribute to elevated apoptosis and may be involved in abnormal oocyte maturation in patients with polycystic ovarian syndrome [44]. In addition, the FKN chemokine and its receptor are overexpressed in endometriosis and may be involved in endometriotic pain conduction [45].

In this study, we focused on the effect of iron deficiency anemia on the secreted cytokines (IL-6, IL-1β, LIF, TGF-β), chemokines (IL-8, FKN), and other regulatory proteins (bone morphogenic protein 2 (BMP2), activin, follistatin, PR, SOX-17, PTGER2, TIMP2) and the modifying effect of FKN on the expression of these proteins, which may improve endometrium receptivity. Based on the findings, it seems that FKN ameliorates the negative effect of iron deficiency on the receptivity-related genes and proteins of HEC-1A endometrium cells, suggesting its important role in the regulation of endometrium receptivity.

## 2. Results

### 2.1. Iron Deficiency Significantly Decreases the mRNA Expression Levels of the Endometrium Receptivity-Related Genes

After the development of anemia in HEC-1A cells, the relative mRNA expression levels of endometrium receptivity-related genes were determined by Real-time PCR. Both the CX3CR1 and PR mRNA levels significantly decreased at 24 h and 48 h compared to the control (Figure 1A). After 72 h long DFO treatment, their mRNA expression levels were elevated to almost double the control level (considered 1) in the case of CX3CR1 and to the control level in the case of PR (Figure 1A), suggesting that possible iron redistribution in the cells may improve the expression rates (Appendix A) and/or that the DFO mediated reduction in FKN expression activates CX3CR1 transcription (Figure 1B).

The mRNA expression levels of FKN regulating CX3CR1 signaling, and SOX-17 controlled by CX3CR1, and PR were significantly downregulated at 24 and 48 h (Figure 1B). Only SOX-17 level increased at 72 h DFO treatment to the control level following the elevation of both CX3CR1 and PR (Figure 1B).

The mRNA levels of the activin receptor activator activin and the inhibitor follistatin showed decreased expression rates (Figure 1C) at 24 h and 48 h. On the contrary, at 72 h DFO treatment, activin mRNA level increased two-fold compared to the control, while follistatin mRNA level increased to the control level (Figure 1C).

The BMP2 mRNA level was reduced during the whole experiment, suggesting that activin and BMP2 reacted differently to iron deficiency (Figure 1C).

We analyzed the mRNA levels of the secreted cytokines involved in the development of endometrium receptivity. IL-1β showed the lowest mRNA levels upon iron deficiency (Figure 1D). LIF and IL-6 mRNA levels behaved similarly at DFO treatment. At 24 h and 48 h, their expression decreased. However, at 72 h, the mRNA levels became elevated near to the control level (Figure 1D). IL-8 and TGF-β also showed similar alterations at iron deficiency. At 24 h, their mRNA expression levels were downregulated, but their levels later began to increase, especially at 72 h, when the IL-8 level reached 1.53 and the TGF-β level was elevated to 1.86 compared to the control.

According to the results, iron deficiency reduces the mRNA expression levels of the examined genes at 24 h and 48 h, which was partially reversed at 72 h. IL-8 and TGF-β mRNA expressions seem to be less sensitive to the iron availability of the HEC-1A cells.

### 2.2. Fractalkine Ameliorates the Effect of Iron Deficiency on the mRNA Expression of Activin and Follistatin, and the Secretions of FKN and BMP2 of the HEC-1A Cells

In the first experiment, when the cells were cultured in a serum-free medium during the entire treatment, the addition of FKN significantly upregulated activin mRNA expression compared to both 24 h and 48 h DFO treatments (Figure 2A). The follistatin mRNA expression was maintained at a low level by FKN similar to the DFO treatment alone (Figure 2A).

In the second experiment, after the DFO treatments, the culture medium was supplemented with serum (containing iron as a transferrin-bound form) together with FKN for 24 h. The presence of serum iron triggered activin mRNA expression in DFO-treated HEC-1A cells, while the addition of 10 ng/mL of FKN significantly elevated the activin mRNA levels compared to DFO treatments (Figure 2B). Interestingly, 20 ng/mL of FKN treatment seemed to be less effective in increasing activin mRNA levels, suggesting a concentration-dependent effect of FKN (Figure 2B). The presence of serum in the culture medium upregulated follistatin mRNA expression to the control level, but FKN treatment did not affect its expression (Figure 2B).

The secreted levels of BMP2 and FKN were also determined after the development of anemia. It was revealed that DFO decreased, while FKN significantly increased BMP2 production of HEC-1A cells in a serum-free environment, suggesting that FKN can modify BMP2 synthesis without the presence of iron (Figure 2C). Using serum and FKN together after DFO treatments, the positive effect of FKN was less obvious after 24 h DFO treatment but was much more effective in elevating BMP2 production after 48 h DFO treatment compared to the serum-free experiment (Figure 2D).

Treatment with 10 ng/mL of FKN was significantly more efficient compared to the higher FKN concentration in both experiments (Figure 2C,D). In the case of FKN secretion, DFO treatments significantly decreased FKN synthesis compared to the control (Figure 2E,F). Meanwhile, FKN supplementation in a serum-free medium increased FKN secretion in a concentration-dependent manner (Figure 2E). When the cells were treated with FKN in a serum-containing culture medium, the elevation of FKN secretion was more pronounced compared to serum-free treatment, showing concentration dependence (Figure 2F). According to the fact that the FKN measurements include the concentrations of the recombinant FKN protein used in the treatments, suggesting the triggering effect of FKN on its synthesis.

Based on these observations, it is supposed that FKN provides a positive effect on activin mRNA expression as well as on BMP2 and FKN secretions that may support endometrium receptivity at iron deficiency. Moreover, it seems that the effect of FKN is strengthened by the presence of serum iron.

### 2.3. Fractalkine Alters the Effect of Iron Deficiency on the Secreted Cytokines Related to Endometrium Receptivity

We measured the secreted cytokine concentrations of the iron-deficient HEC-1A cells after FKN treatments cultured with or without serum supplementation. It was found that serum supplementation decreased the LIF, IL-6, and TGF-β secretions in general in the untreated cells as well as in the differently treated cells compared to the cells cultured in serum-free medium. In the case of LIF, no significant change was found after DFO treatments, in the serum-free experiment. Nevertheless, FKN increased LIF synthesis after DFO treatments, especially at 48 h in a concentration-dependent way (Figure 3A). This elevation was only found after 48 h DFO treatment when FKN was added together with serum-containing medium (Figure 3B).

IL-6 secretion was reduced after 24 h DFO treatment and was significantly upregulated using FKN supplementation in a concentration-dependent manner (Figure 3C). This observation suggests that FKN activates IL-6 synthesis via the CX3CR1 pathway.

Interestingly, DFO treatment did not decrease IL-8 secretion of HEC-1A cells, as it did in the case of LIF and IL-6 (Figure 3A,C,E). Moreover, DFO and FKN increased IL-8 synthesis at the same level, suggesting that FKN did not affect IL-8 secretion in iron deficiency (Figure 3E,F).

In the case of TGF-β, iron deficiency elevated the TGF-β secretion rather than reduce it, but in contrast with IL-8, FKN significantly increased the TGF-β protein levels compared to the DFO treatments (Figure 3G,H). We must note that the supplementation of the culture medium with serum after DFO treatments drastically reduced the TGF-β secretion compared to the serum-free experiments, but FKN was still able to significantly elevate the TGF-β synthesis of HEC-1A cells (Figure 3G,H).

TNFα level was not detected after 24 h DFO treatment, but FKN significantly increased its level compared to DFO treatment alone. Although FKN mediated TNFα secretion did not reach the control level (Figure 3I), using serum supplementation for the last 24 h of the experiment increased the TNFα levels, suggesting that the reintroduction of iron affects TNFα production of the HEC-1A cells. FKN addition after both DFO treatments elevated TNFα levels (Figure 3J). Interestingly, after 24 h DFO treatment, the TNFα secretion was significantly higher using 20 ng/mL of FKN compared to the effect of 10 ng/mL FKN. Meanwhile, after 48 h of DFO treatment, the TNFα secretion was significantly lower using 20 ng/mL of FKN compared to the effect of 10 ng/mL FKN (Figure 3J).

Our findings support the hypothesis that FKN triggers the secretion of the examined receptivity-related cytokines present in the case of anemia (except IL-8) and may improve endometrium receptivity.

### 2.4. Fractalkine Modifies the Levels of the Endometrium Receptivity-Related Proteins at Iron Deficiency

Next, the protein levels of PR, P-PR, CX3CR1, and the downstream signaling molecules, SOX-17, Nrf2, and Keap-1 transcription factors, were determined. The level of PR decreased due to iron deficiency at 24 h, even if serum was added to the cells for 24 h after DFO treatments (Figure 4A–C). After 24 h long DFO treatment, the supplementation of FKN significantly increased PR protein expression. At 24 h, using 20 ng/mL of FKN exerted a stronger effect compared to the lower FKN concentration (Figure 4A–C). At 24 h, the phosphorylation of PR followed the increasing level of PR (Figure 4B,C). We have to note that the presence of serum during the last 24 h of the experiment elevated the PR protein levels compared to the serum-free experiments, suggesting that iron (or maybe other factors) contributes to the PR synthesis.

PR phosphorylation can be increased by CX3CR1/FKN signaling [33]. Interestingly, in the serum-free experiment, 20 ng/mL of FKN decreased the CX3CR1 protein level, while the serum supplementation reversed this effect, and significantly increased CX3CR1 expression at 24 h (Figure 4A,D).

The expression of SOX-17 is indirectly regulated by CX3CR1 via the phosphorylation of PR. The SOX-17 protein level decreased only after 24 h of DFO treatment compared to the control, while FKN administration significantly increased the SOX-17 level compared to the DFO treatment (Figure 4A,E). Supplementation with serum at FKN treatment changed the effect of FKN on the iron-deficient HEC-1A cells, significantly reducing the SOX-17 level (Figure 4A,E).

The Nrf2/Keap-1 system is regulated by the CX3CR1 and PR signaling pathways, but it also functions as the controller of CX3CR1-mediated signaling (e.g., NFκB), and therefore the cytokine production of the cells [36,37]. At 24 h, even if the iron source (serum) was readded to the cells, DFO treatment decreased the Nrf2 level (Figure 4A,F). On the other hand, the presence of iron modified the action of FKN on Nrf2 expression. In the serum-free experiment, the 10 ng/mL FKN treatment decreased, while the 20 ng/mL of FKN increased the Nrf2 protein level compared to 24 h DFO treatment (Figure 4F). In the case of the Nrf2 inhibitor Keap-1, iron deficiency elevated its protein level, while utilization of FKN at the higher concentration significantly decreased Keap-1 expression (Figure 4G).

The level of PR decreased due to iron deficiency at 48 h, even if serum was added to the cells for 24 h after DFO treatments (Figure 5A,B). At 48 h, the level of the phosphorylated PR reached only the control level (Figure 5A,C). However, the upregulating effect of FKN was significant compared to the DFO treatment (Figure 5C). Serum supplementation affected CX3CR1 level neither at iron deficiency nor FKN administration at 48 h (Figure 5A,D).

Considering the 48 h DFO treatments, iron deficiency increased SOX-17 protein level, and FKN reduced its level at both concentrations (Figure 5A,E). Using a serum-containing medium at FKN administration, FKN acted the same way, significantly decreasing SOX-17 protein expression compared to the DFO treatment (Figure 5A,E). These observations suggest that SOX-17 synthesis hinges on the alterations of iron homeostasis, too.

In the serum-free experiment, the 10 ng/mL FKN treatment decreased, while the 20 ng/mL of FKN increased the Nrf2 protein level compared to 48 h DFO treatment (Figure 5F). The administration of serum with FKN diminished the effect of FKN on the Nrf2 protein synthesis, suggesting that iron contributes to the regulation of Nrf2 (Figure 5G). Similar to the 24 h long treatments, in the case of the Nrf2 inhibitor Keap-1, iron deficiency elevated its protein level, while utilization of FKN at the higher concentration significantly decreased Keap-1 expression at 48 h (Figure 5A,G). Based on the results, the reintroduction of iron to the cells by serum supplementation affects the level of Keap-1 predisposing a link between iron availability and Keap-1 synthesis.

### 2.5. The Effect of Fractalkine on the Expression of PTGER2 and TIMP2 Depends on the Iron Availability of HEC-1A Cells

PTGER2 and TIMP2 proteins are also involved in endometrium receptivity and the regulation of embryo implantation [22,24]. Short-term iron deficiency reduced the PTGER2 level, and FKN was able to increase its expression at both concentrations without the presence of iron (Figure 6A,C). When FKN was added to the cells together with serum supplementation, its effect was reversed, and FKN reduced the PTGER2 protein expression of the HEC-1A cells (Figure 6A,C). In the case of 48 h long iron deficiency, DFO treatment elevated PTGER2 level, and administration of FKN inhibited the effect of DFO (Figure 6B,E). Interestingly, when serum was added to the DFO-treated cells for the last 24 h of the experiment, the PTGER2 protein level significantly decreased compared to the control (Figure 6B,E). FKN addition to the cells in the presence of serum further reduced the PTGER2 level (Figure 6B,E).

DFO treatment significantly reduced TIMP2 protein levels in each type of experiment (Figure 6A,B,D,F). The 10 ng/mL of FKN was only successful in increasing TIMP2 protein expression after 24 h DFO treatment in a serum-free environment (Figure 6A,D). In all the other cases FKN at lower concentration decreased TIMP2 level compared to the DFO treatment (Figure 6A,B,F). The 20 ng/mL of FKN was only able to raise the TIMP2 level in the serum-free experiments during both periods (Figure 6D,F). In the case of serum supplementation after DFO treatments, FKN (20 ng/mL) significantly elevated TIMP2 protein expression compared to 10 ng/mL FKN treatment but not compared to DFO treatment (Figure 6D,F).

These findings presuppose that fractalkine acts against anemia by increasing the protein synthesis of PTGER2 and TIMP2, but the addition of serum (iron) reverses its effect.

## 3. Discussion

Fractalkine (CX3CL1/FKN) is the sole chemokine belonging to the CX3C chemokine subclass [46]. The membrane-bound FKN has an extended C terminus containing a mucin-like stalk and a transmembrane domain [47]. FKN is expressed by various cell types, e.g., neurons, astrocytes, endometrium cells, trophoblasts, endothelial cells, and immune cells [48,49,50]. The soluble form of FKN is produced by the cleavage of the membrane-bound FKN mediated by ADAM proteins [51,52]. Soluble FKN mainly acts as a chemoattractant molecule [52]. Both types of FKN can bind to and activate their receptor, the CX3CR1. CX3CR1, a G protein-coupled receptor, is expressed by microglia, macrophages, T cells, smooth muscle cells, endometrium cells, and trophoblasts [47,48,53,54]. The FKN/CX3CR1 axis regulates three signaling pathways, the NFκB, MAPK and PKC, which act on the cell cycle and proliferation, inflammation and secretion processes [55].

FKN is implicated in maternal–fetal communication and regulates implantation-related genes, and endometrial receptivity as well [33,34,49]. Our research group revealed that FKN affects the iron metabolism of the endometrium cells and enhances iron transport towards the trophoblast cells surrounding the embryo [34]. It has been also described that iron, namely the iron status of the mother, is essential in fertility. Therefore, iron availability in the embryo contributes to fertilization and pregnancy [7,8,9]. Anemia may reduce conception and causes infertility [8,56].

In the present study, we focused on the action of FKN on the endometrium receptivity-related genes and proteins to reveal if FKN has a beneficial effect to ameliorate the outcome of iron deficiency in HEC-1A cells. To reveal whether the effect of FKN is iron-dependent, the experiments were carried out in a serum-free and a serum-supplemented version. In the latter case, FKN was added together with serum for the last 24 h of the experiment modelling transferrin-bound iron supplementation.

The mRNA expression of the endometrium receptivity-related genes, PR, SOX-17, CX3CR1, FKN, activin, follistatin, BMP2, as well as the examined cytokines and chemokines LIF, IL-6, and IL-1β, were significantly downregulated at DFO-induced iron deficiency in HEC-1A cells. Only IL-8 and TGF-β were found to be less sensitive to iron depletion. These results predispose the essential role of iron in the regulation of receptivity-related genes.

Endometrium receptivity is a complex process regulated by a large number of proteins [3]. Cytokines, such as TNFα and IL-1β, secreted by the endometrium initiate an inflammatory process. The elevated levels of these cytokines, as well as IL-6 and IL-8, help in the process of embryo implantation [15,24,57,58]. The synthesis of the inflammatory mediators is under the regulation of the NFκB signaling pathway, which can be triggered by the CX3CR1/FKN axis [59]. The NFκB signaling pathway can be modulated by the cytokines themselves, e.g., TNFα, IL-1β, or by the Nfr2/Keap-1 transcriptional regulators [60]. In our experiments, the absence of iron decreased LIF, IL-6, and TNFα secretions of the HEC-1A cells. Meanwhile, using DFO for 48 h preferably increased cytokine expression compared to the controls. Moreover, in the case of IL-8 and TGF-β, iron deficiency elevated their levels, suggesting the distinct roles of iron deficiency in the modulation of cytokine secretion. The other possible reason for this observation could be the decreasing level of FKN and consecutive downregulation of the activation of CX3R1 and NFκB signaling. The treatments with the two different concentrations of FKN, increased the secretion of all five cytokines in a concentration-dependent manner, suggesting the reactivation of the CX3CR1 receptor. These changes were less dramatic after the serum supplementation at FKN treatments in the case of LIF, IL-6, and TGF-β, but the TNFα production was revealed to be significantly elevated compared to the serum-free experiment. This interesting result may support that FKN functions as a regulator of cytokine production, although iron can modify its action. Only the IL-8 chemokine level showed no alteration after FKN treatments. One possible explanation for this result is that the mRNA of IL-8 is very unstable, and therefore less IL-8 protein can be translated from the mRNA molecules [61]. Another reason for the unaltered IL-8 level is its function. IL-8 is involved in trophoblast invasion and does not have a crucial role in endometrium receptivity [27].

Activins are also secreted by endometrial cells, and they are involved in the progress of receptivity [62,63]. Activins are inhibited by follistatin, which irreversibly binds to activins and inhibits receptor activation [64]. FKN increased activin mRNA synthesis at iron-deficient HEC-1A cells and decreased or maintained the control level of follistatin contributing to the action of activin on its receptor, which regulates cell division and differentiation [65]. BMP2 is also implicated in endometrium receptivity and fertility [66,67]. Recently, it has been discovered that endometrium receptivity requires BMP2/activin receptor type 2A (ACVR2A) signaling [68]. In our study, FKN significantly increased BMP2 production of the iron-deficient HEC-1A cells both in serum-free and serum-supplemented FKN treatments. This observation proposes that FKN may contribute to BMP2/ACVR2A signaling.

Based on the results, it seems that iron deficiency reduces FKN synthesis of the HEC-1A cells, and it cannot be improved by serum (iron) supplementation suggesting that FKN production does not hinge on the intracellular iron content. We also demonstrated that FKN acts in an autocrine way on its expression and further triggers FKN secretion in a concentration-dependent way. The elevated level of FKN due to FKN treatment of the iron-deficient cells may contribute to the increased activity of CX3CR1 and the downstream signaling pathways.

In accordance with these findings, the protein levels of the CX3CR1 as well as the P-PR, SOX-17, and the Nrf2/Keap-1 system were determined. The CX3CR1 level only decreased after the addition of 20 ng/mL of FKN. Otherwise, its level was upregulated by FKN, or did not change, suggesting that the signaling via CX3CR1 is active. The phosphorylation of PR is regulated by the MAPK pathway controlled by CX3CR1 [69]. We found that FKN significantly increased P-PR level at iron deficiency, suggesting the improving effect of FKN on PR activation. PR and its mediators are crucial in the development of the window of receptivity [70]. One of the target genes of P-PR is SOX-17, whose elevated level is pivotal in endometrium receptivity [40,71]. Although our previous study revealed that SOX-17 expression is operated by CX3CR1-mediated PR activation, the action of FKN on SOX-17 protein expression is controversial in anemia, probably due to the altered level of progesterone in a serum-free medium. After shorter iron deficiency, FKN increased the SOX-17 level, but after 48 h it was not capable to maintain SOX-17 protein synthesis. When serum was added together with FKN, the SOX-17 level was lower compared to the control. After 48 h long iron deficiency, FKN in the presence of serum significantly reduced SOX-17 protein level. These results suggest that when the cells obtain iron and progesterone together, FKN prevents the cells from overproduction of SOX-17.

The other target of P-PR is the Nrf2/Keap-1 transcription factor system, in which Keap-1 inhibits the action of Nrf2 [38]. The Nrf2 transcription is indirectly regulated by CX3CR1 via PR. Nrf2 modifies the activity of the NFκB signaling pathway regulating cytokine transcription involved in the development of endometrium receptivity [37]. In our experiments, Keap-1 synthesis was inhibited by FKN treatments, suggesting the activation of Nrf2. The Nrf2 level was raised using the higher concentration of FKN in a serum-free environment. The presence of iron and other factors, e.g., progesterone, decreased the elevating effect of FKN on Nrf2.

IL-1β and the NFκB pathway indirectly activate prostaglandin synthesis by upregulating the transcription of prostaglandin E synthase [72]. Moreover, LIF also contributes to the production of prostaglandins [73], triggering the activity of PTGER2. PTGER2 regulates endometrium receptivity by modifying the expression of tissue inhibitors of metalloproteinases (TIMPs) [22,24]. In the HEC-1A cells, the PTGER2 and TIMP2 levels showed a parallel alteration. Iron deficiency decreased their levels at 24 h, while FKN addition reversed this effect, but later iron deficiency increased PTGER2 and decreased TIMP2, FKN in turn decreased PTGER2 and increased TIMP2 compared to DFO treatments. The effect of FKN predisposes its role in regulating the balance in prostaglandin synthesis at anemia in HEC-1A cells. Interestingly, the addition of serum together with FKN reverses its effect on TIMP2 protein, which may enhance implantation.

Our findings support the hypothesis that FKN ameliorates the effects of iron deficiency on the receptivity-related genes and proteins in HEC-1A cells. FKN triggers the secretion of the receptivity-related cytokines at anemia that may improve endometrium receptivity via the CX3CR1 and the regulation of the NFκB pathway. FKN can contribute to cell proliferation and differentiation by regulating activin, follistatin, and BMP2. FKN also alters the protein levels of PR, SOX-17, PTGER2, and TIMP2, which may contribute to embryo implantation. FKN indirectly modifies the Nrf2/Keap-1 system, contributing to the regulation of the NFκB pathway controlling cytokine transcription (Figure 7).

The effect of FKN seems to be iron independent in that it can upregulate the expression of the target proteins. However, serum supplementation alters the outcome of FKN treatment. Hence, we cannot exclude that not only iron, but also additional serum components are responsible for these differences. Although these findings provide a deeper insight into the regulation of receptivity-related genes and proteins, the major limitation of the study is the utilization of the in vitro cell culture model. The results obtained from our studies need to be verified in an in vivo animal or a human study.

## 4. Materials and Methods

### 4.1. Cell Culture and Treatments

HEC-1A cells (ATCC HTB-112) were cultured in McCoy’s 5A Iwakata and Grace modification (Corning Ltd., New York, NY, USA) supplemented with 10% fetal bovine serum (FBS, EuroClone S.p.A., Pero, Italy) and 1% penicillin/streptomycin (P/S 10k/10k, Lonza Ltd., Basel, Switzerland) in a humified atmosphere containing 5% CO_2_ at 37 °C. For the Real-time PCR, 5 × 10^5^ cells were plated onto 6-well dishes (Biologix Europe Ltd., Hallbergmoos, Germany), and for the Western blot experiments, 10^6^ cells were cultured on 6 cm Petri dishes (Biologix Europe Ltd., Hallbergmoos, Germany). The development of iron deficiency was proven by the decreasing mRNA level of ferritin heavy chain (FTH; Appendix A) after administering 10, 25, 50, 100, and 500 µM desferrioxamine (DFO; Novartis Hungária Kft., Budapest, Hungary) for 24 h, 48 h, and 72 h in a serum-free culture medium. The 100 µM DFO concentration was chosen according to the results of dose-dependence and viability experiments (Appendix A). The untreated cells were incubated for the same time (24 h, 48 h, and 72 h) in a serum-free environment as the DFO-treated cells and were used as controls at the corresponding time point. For the determination of the effect of FKN on the endometrium receptivity-related genes and proteins, the following treatments were used: 1. The cells were treated with DFO for 24 h or 48 h in a serum-free culture medium. Then, 10 ng/mL or 20 ng/mL of human recombinant FKN (Biolegend, San Diego, CA, USA) were added to the cultures for an additional 24 h (the whole experiment was carried out in a serum-free environment). 2. The cells were treated with DFO for 24 h or 48 h in serum-free culture medium. Then, the culture medium was changed to serum-containing medium supplemented with 10 ng/mL or 20 ng/mL of human recombinant FKN for an additional 24 h. In the experiments, the serum-free medium contributes to iron deficiency (iron is not available for the cells), and the serum-containing medium provides iron available again for the cells in a transferrin-bound form. DFO and FKN were solved in sterile distilled water (100 mM DFO; 0.1 mg/mL FKN). The untreated control cells were supplemented with the same volume of solvent in the same way used in the treated cells.

### 4.2. Real-Time PCR

The cells were collected after the treatments, and total RNA was isolated using Aurum Total RNA Mini Kit (Bio-Rad Inc., Hercules, CA, USA) according to the manufacturer’s protocol. RNA concentration of the samples was measured by MultiSkan GO spectrophotometer (Thermo Fisher Scientific Inc., Waltham, MA, USA) using the µDrop plate (Thermo Fisher Scientific Inc., Waltham, MA, USA). Complementary DNA was synthesized using 200 ng of total RNA from each sample by iScript Select cDNA Synthesis Kit (Bio-Rad Inc., Hercules, CA, USA). Gene-specific primers were designed using Primer-BLAST [74] and were purchased from Integrated DNA Technologies (IDT, Leuven, Belgium). The primer sequences can be found in Table 1. Real-time PCR was performed in a CFX96 One Touch Real-Time PCR System (Bio-Rad Inc., Hercules, CA, USA) using SYBR Green protocol and iTaq Universal SYBR Green Reagent Mix (Bio-Rad Inc., Hercules, CA, USA). After each run, a melting curve was generated to prove that only one product was amplified in each tube. Relative gene expression was calculated by the Livak (ΔΔCt) method using glyceraldehyde 3-phosphate dehydrogenase (GAPDH) housekeeping gene for normalization and Bio-Rad CFX Maestro 3.1. software (Bio-Rad Inc., Hercules, CA, USA). Relative gene expression was expressed as fold change. The target genes’ relative expression was considered one in the controls.

### 4.3. Enzyme-Linked Immunosorbent Assay (ELISA) Measurements

The supernatants of the treated and control HEC-1A cells were collected and stored at −80 °C until the ELISA measurements. The secreted LIF, BMP2, and TGF-β protein concentrations were determined with ELISA kits purchased from Merck Kft. (Merck Life Sciences Kft., Budapest, Hungary) according to the manufacturer’s protocols. The concentration of the secreted IL-6, IL-8, TNFα, and FKN was measured using ELISA kits purchased from Thermo Fisher Scientific Inc. (Thermo Fisher Scientific Inc., Waltham, MA, USA) following the instructions of the manufacturer. FKN ELISA kit detects both soluble and recombinant forms of FKN protein. All measurements were performed in triplicate.

### 4.4. Western Blot Analysis

For the protein expression analysis, the differently treated and control cells were collected by centrifugation. The cell pellets were lysed with 150 µL of ice-cold lysis buffer (50 mM Tris-HCl, pH 7.4, 150 mM NaCl, 0.5% Triton-X 100) containing a complete mini protease inhibitor cocktail (Roche Ltd., Basel, Switzerland). The protein content of each sample was determined using the DC Protein Assay Kit (Bio-Rad Laboratories, Hercules, CA, USA) and MultiSkan GO spectrophotometer (Thermo Fisher Scientific Inc., Waltham, MA, USA). An equal amount of protein from each sample was separated on 10% (PR, P-PR, Nrf2, and Keap-1) or 12% (CX3CR1, SOX-17, PTGER2 and TIMP2) polyacrylamide gels. The Mini Protean Tetra Cell equipment (Bio-Rad Laboratories, Hercules, CA, USA) was used for electrophoresis. The protein-containing gels were transferred by electroblotting to nitrocellulose membranes (Pall AG, Basel, Switzerland) and were blocked with 5% (*w*/*v*) non-fat dry milk in TBST for 1 h at room temperature with gentle shaking. The membranes were probed with the following antibodies for 1 h at room temperature: anti-progesterone receptor (PR) IgG (1:1000; Thermo Fisher Scientific Inc., Waltham, MA, USA, cat.no.: PA5-17180), anti-phospho-PR IgG (1:1000; Thermo Fisher Scientific Inc., Waltham, MA, USA, cat.no.: PA5-104635), anti-fractalkine receptor (CX3CR1) IgG (1:1000; Thermo Fisher Scientific Inc., Waltham, MA, USA, cat.no.: PA1-12541), anti-SOX-17 IgG (1:1000; Merck Life Sciences Kft., Budapest, Hungary, cat.no.: 09-038-I), anti-prostaglandin receptor E2 (PTGER2) IgG (1:1000; Thermo Fisher Scientific Inc., Waltham, MA, USA, cat.no.: bs-4196R), and anti-TIMP2 IgG (1:1000; Thermo Fisher Scientific Inc., Waltham, MA, USA, cat.no.: PA1-25313). In the case of Nrf2 IgG (1:1000, Cell Signaling Technology Europe, Leiden, The Netherlands, cat.no.: 12721S) and anti-Keap-1 IgG (1:1000; Cell Signaling Technology Europe, Leiden, The Netherlands, cat.no.: 4678S), overnight incubation at 4 °C was conducted. GAPDH (anti-GAPDH IgG, 1:3000; Merck Life Science Kft., Budapest, Hungary, cat.no.: G9545) was used as a loading control for the Western blots. For the secondary antibody, goat anti-rabbit IgG (H + L) HRP conjugated IgG was used (1:3000; Merck Life Science Kft., Budapest, Hungary, cat.no.: AP307P) for 1 h at room temperature. The development of WBs was carried out in a UVItec Alliance Q9 Advanced imaging system (UVItec Cambridge Ltd., Cambridge, UK) using WesternBright ECL chemiluminescent substrate (Advansta Inc., San Jose, CA, USA). The optical density of the protein bands was determined by ImageJ software [75]. The protein levels were expressed as a percentage of the target protein/GAPDH ratio.

### 4.5. Data Analysis

The real-time PCR analyses and the ELISA measurements were carried out in triplicate in three independent experiments. Western blots shown in the figures are representative of three independent experiments. Data are shown as the mean ± standard deviation (SD). Statistical analysis was performed using SPSS software version 24.0 (IBM Corporation, Armonk, NY, USA). Statistical significance was determined by two-way ANOVA followed by Scheffe’s post hoc test. The results were regarded statistically significant if the *p*-value was <0.05.

## 5. Conclusions

Iron deficiency anemia is a global health problem affecting a remarkable fraction of the fertile female population. The prevalence of anemia in pregnant women is reasonably high. Maternal iron metabolism plays a crucial role in the iron supplementation of the growing fetus, it affects embryonic development in many aspects e.g., neurodevelopment. It has been proven that the chemokine FKN plays a role in endometrium receptivity and embryo implantation. Moreover, FKN may function as a regulator of cellular iron metabolism. According to our study, FKN may function as a positive regulator of the expression of receptivity-related genes and proteins at iron deficiency. FKN induces the secretion of the receptivity-related cytokines at anemia, which may improve endometrium receptivity via the CX3CR1 and the regulation of the NFκB pathway and by regulating activin, follistatin, and BMP2, as well as PR, SOX-17, PTGER2, and TIMP2. FKN level might contribute to the development of endometrium receptivity and successful embryo implantation.

## Figures and Tables

**Figure 1 ijms-24-07924-f001:**
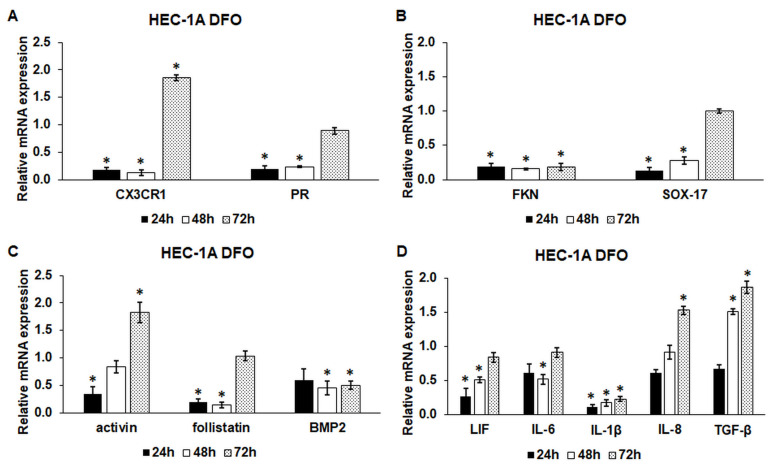
Real-time PCR analysis of the mRNA expression of the endometrium receptivity-related genes after 24 h, 48 h and 72 h long DFO treatments. DFO treatments were carried out in a serum-free culture medium. Real-time PCR was performed using an SYBR green protocol. For the normalization of the gene expression levels, GAPDH was used as a housekeeping gene. The untreated cells were used as a control in the experiment. (**A**) mRNA expression levels of fractalkine receptor (CX3CR1) and progesterone receptor (PR). (**B**) Expression levels of fractalkine (FKN) and SOX-17. (**C**) Relative mRNA levels of activin, follistatin and BMP2. (**D**) Expression levels of cytokines, LIF, IL-6, IL-1β, IL-8 and TGF-β. The relative expression level of the target genes of the control was regarded as 1. The columns represent the mean ± SD of three independent experiments (*n* = 3). The analysis was carried out in triplicate/sample in each experiment. The asterisk shows *p* < 0.05 compared to the control.

**Figure 2 ijms-24-07924-f002:**
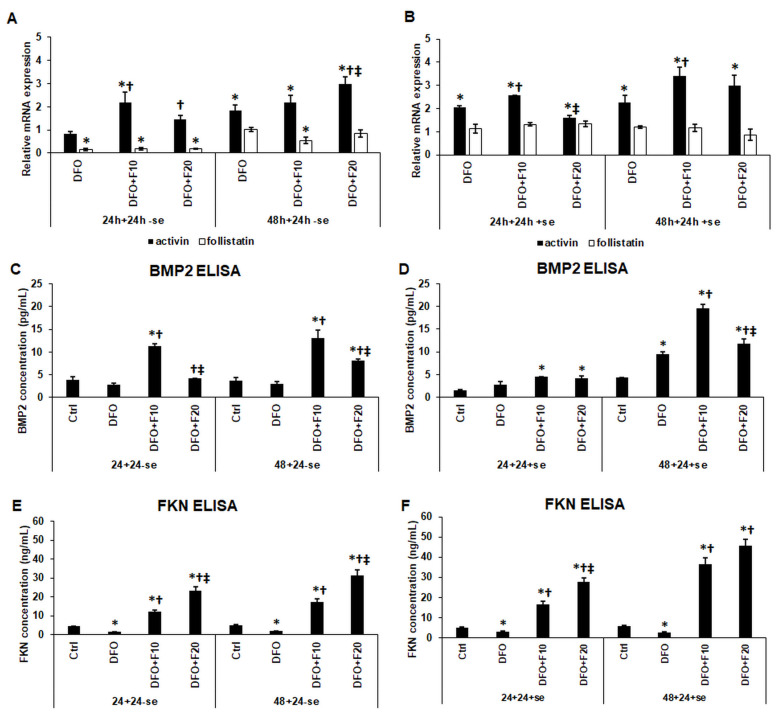
mRNA expression analysis and ELISA measurements of the endometrium receptivity-related genes and secreted proteins after 24 h and 48 h long DFO treatments in a serum-free environment followed by 24 h fractalkine supplementation in serum-free and serum-containing culture media. Real-time PCR was performed using an SYBR green protocol. For the normalization of the gene expression levels, GAPDH was used as a housekeeping gene. The untreated cells were used as a control in the experiment. The relative expression level of the target genes of the control was regarded as 1. ELISA measurements were carried out using BMP2 and FKN specific human ELISA kits following the manufacturer’s protocols. (**A**) mRNA levels of activin and follistatin after 24 h and 48 h DFO treatment followed by 24 h FKN supplementation in a serum-free environment. (**B**) mRNA levels of activin and follistatin after 24 h and 48 h DFO treatment followed by 24 h FKN supplementation in a serum-containing environment. (**C**) Secreted BMP2 concentration after 24 h and 48 h DFO treatment followed by 24 h FKN supplementation in a serum-free culture medium. (**D**) Secreted BMP2 concentration after 24 h and 48 h DFO treatment followed by 24 h FKN supplementation in a serum-containing culture medium. (**E**) Secreted FKN concentration after 24 h and 48 h DFO treatment followed by 24 h FKN supplementation in a serum-free culture medium. (**F**) Secreted FKN concentration after 24 h and 48 h DFO treatment followed by FKN supplementation in a serum-containing culture medium. The columns represent the mean ± SD of three independent experiments (*n* = 3). The determinations were carried out in triplicate/sample in each experiment. The asterisk shows *p* < 0.05 compared to the control. The cross means *p* < 0.05 compared to the DFO treatments. The double cross signs *p* < 0.05 compared to the F10 treatment. Abbreviations: DFO-desferrioxamine; FKN-fractalkine; F10-fractalkine 10 ng/mL, F20-fractalkine 20 ng/mL.

**Figure 3 ijms-24-07924-f003:**
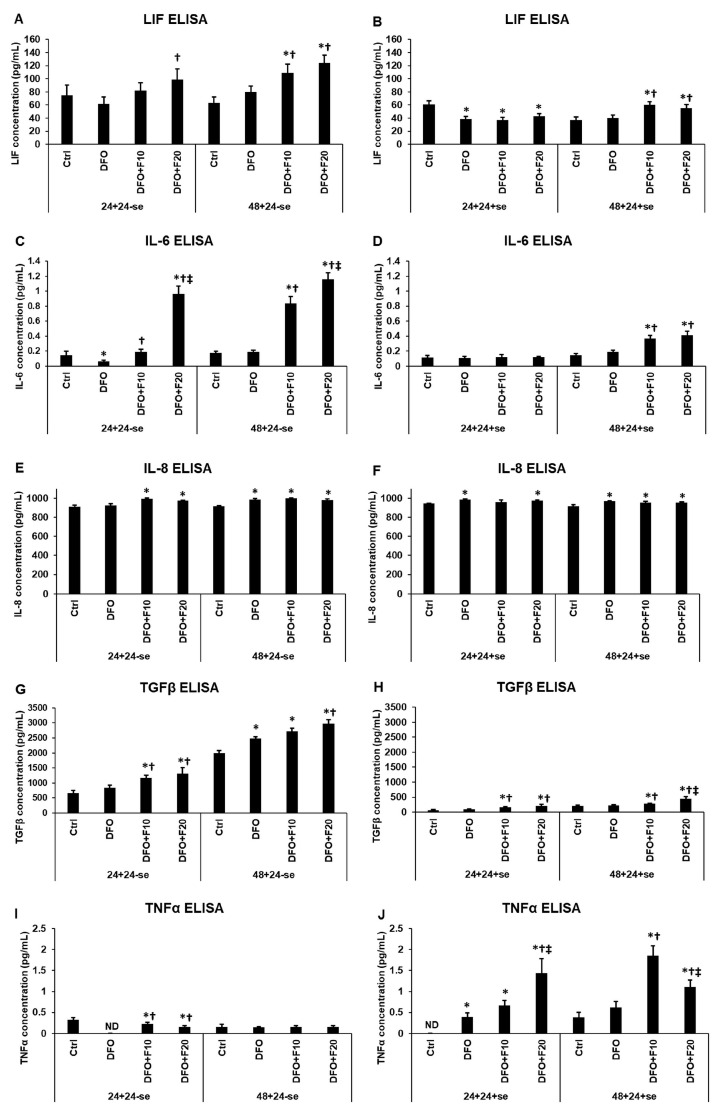
ELISA measurements of the secreted cytokines involved in the endometrium receptivity after 24 h and 48 h long DFO treatments in a serum-free environment followed by 24 h fractalkine supplementation in serum-free and serum-containing culture media. ELISA measurements were carried out using human LIF, IL-6, IL-β, IL-8 and TGF-β specific ELISA kits following the manufacturer’s protocols. (**A**) LIF concentration after 24 h and 48 h DFO treatment followed by 24 h FKN supplementation in a serum-free culture medium. (**B**) LIF concentration after 24 h and 48 h DFO treatment followed by 24 h FKN supplementation in a serum-supplemented culture medium. (**C**) Secreted IL-6 concentration after 24 h and 48 h DFO treatment followed by 24 h FKN supplementation in a serum-free culture medium. (**D**) Secreted IL-6 concentration after 24 h and 48 h DFO treatment followed by 24 h FKN supplementation in a serum-supplemented culture medium. (**E**) Secreted IL-8 concentration after 24 h and 48 h DFO treatment followed by 24 h FKN supplementation in a serum-free culture medium. (**F**) Secreted IL-8 concentration after 24 h and 48 h DFO treatment followed by 24 h FKN supplementation in a serum-supplemented culture medium. (**G**) Secreted TGF-β concentration after 24 h and 48 h DFO treatment followed by 24 h FKN supplementation in a serum-free culture medium. (**H**) Secreted TGF-β concentration after 24 h and 48 h DFO treatment followed by 24 h FKN supplementation in a serum-supplemented culture medium. (**I**) Secreted TNFα concentration after 24 h and 48 h DFO treatment followed by 24 h FKN supplementation in a serum-free culture medium. (**J**) Secreted TNFα concentration after 24 h and 48 h DFO treatment followed by 24 h FKN supplementation in a serum-supplemented culture medium. The columns represent the mean ± SD of three independent experiments (*n* = 3). The measurements were carried out in triplicate/sample in each experiment. The asterisk shows *p* < 0.05 compared to the control. The cross means *p* < 0.05 compared to the DFO treatments. The double cross signs *p* < 0.05 compared to the F10 treatment. Abbreviations: DFO-desferrioxamine; FKN- fractalkine; F10-fractalkine 10 ng/mL, F20-fractalkine 20 ng/mL.

**Figure 4 ijms-24-07924-f004:**
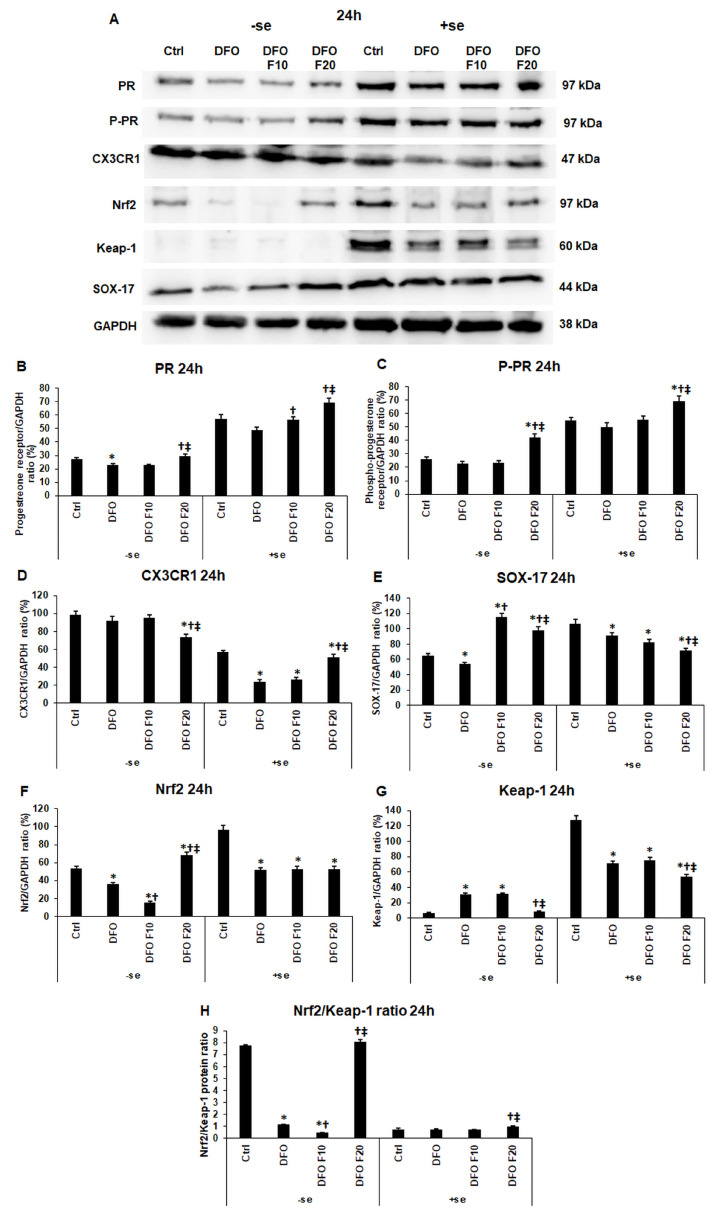
Determination of the expression levels of proteins related to the endometrium receptivity (**A**) after 24 h DFO treatment in a serum-free environment followed by 24 h FKN addition in a serum-free or serum-supplemented culture medium. For the WBs, the HEC-1A cells were harvested after the treatments, then cells were lysed, and the same amount of protein (10 µg) from each sample was separated by SDS-PAGE using 10% polyacrylamide gel. After blotting to nitrocellulose membranes, the membranes were probed with anti-PR, anti-P-PR, anti-CX3CR1, anti-Nrf2, anti-Keap-1 and anti-SOX-17 according to the manufacturer’s instruction. The experiments were repeated three times. GAPDH was used as the loading control. Analysis of the WBs was performed using ImageJ Software version IJ153. (**B**–**G**) Optical density analyses of PR, P-PR, CX3CR1, Nrf2 and Keap-1 in HEC-1A cells after 24 h DFO treatment followed by 24 h FKN addition in a serum-free or serum-supplemented culture medium. (**H**) The protein ratio of Nrf2 and Keap-1. The blots were cropped according to the molecular weight of the target protein. The original blots can be found in the Appendix A. The columns represent the mean ± SD of three independent experiments (*n* = 3). The asterisk shows *p* < 0.05 compared to the control. The cross means *p* < 0.05 compared to the DFO treatments. The double cross signs *p* < 0.05 compared to the F10 treatment. Abbreviations: DFO-desferrioxamine; FKN-fractalkine; F10-fractalkine 10 ng/mL, F20-fractalkine 20 ng/mL.

**Figure 5 ijms-24-07924-f005:**
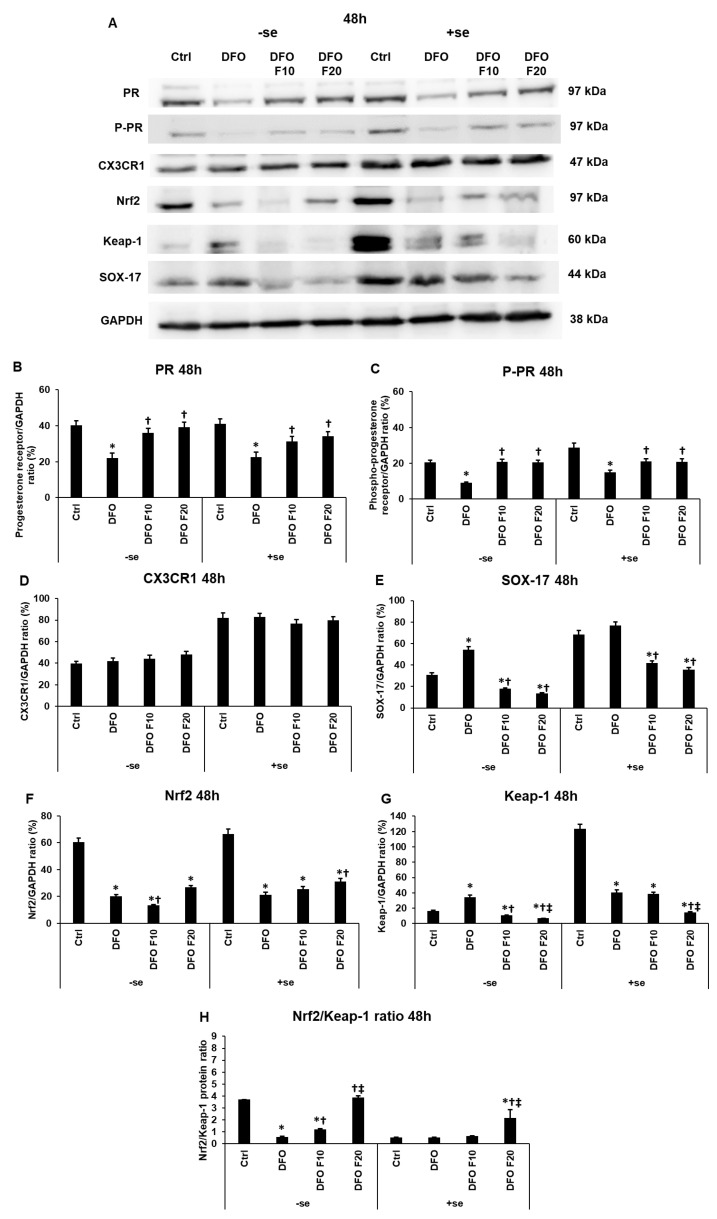
Determination of the expression levels of proteins related to the endometrium receptivity (**A**) after 48 h DFO treatment followed by 24 h FKN addition in a serum-free or serum-supplemented culture medium. For the WBs, the HEC-1A cells were harvested after the treatments, then cells were lysed, and the same amount of protein (10 µg) from each sample was separated by SDS-PAGE using 10% polyacrylamide gel. After blotting to nitrocellulose membranes, the membranes were probed with anti-PR, anti-P-PR, anti-CX3CR1, anti-Nrf2, anti-Keap-1 and anti-SOX-17 according to the manufacturer’s instruction. The experiments were repeated three times. GAPDH was used as the loading control. Analysis of the WBs was performed using ImageJ Software. (**B**–**G**) Optical density analyses of PR, P-PR, CX3CR1, Nrf2 and Keap-1 in HEC-1A cells after 48 h DFO treatment followed by 24 h FKN addition in a serum-free or serum-supplemented culture medium. (**H**) The protein ratio of Nrf2 and Keap-1. The blots were cropped according to the molecular weight of the target protein. The original blots can be found in the Appendix A. The columns represent the mean ± SD of three independent experiments (*n* = 3). The asterisk shows *p* < 0.05 compared to the control. The cross means *p* < 0.05 compared to the DFO treatments. The double cross signs *p* < 0.05 compared to the F10 treatment. Abbreviations: DFO-desferrioxamine; FKN-fractalkine; F10-fractalkine 10 ng/mL, F20-fractalkine 20 ng/mL.

**Figure 6 ijms-24-07924-f006:**
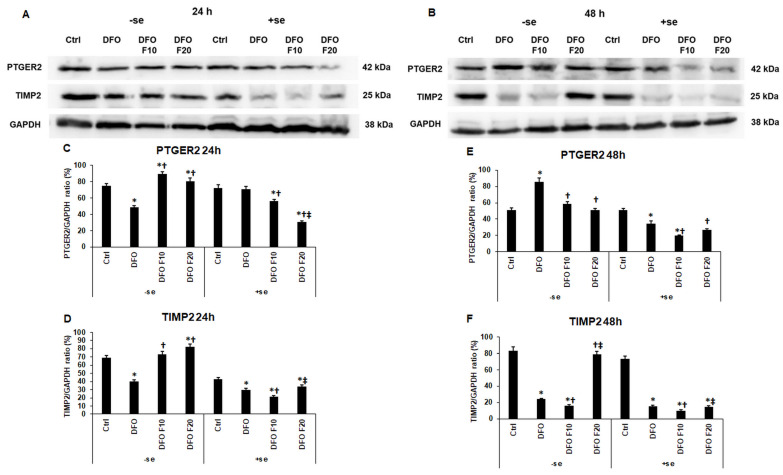
Western blot analysis of PTGER2 and TIMP2 in HEC-1A cells (**A**) after 24 h DFO treatment in a serum-free environment followed by 24 h FKN addition in a serum-free or serum-supplemented culture medium and (**B**) after 48 h DFO treatment in a serum-free environment followed by 24 h FKN addition in a serum-free or serum-supplemented culture medium. After the treatments, the cells were collected and lysed, and then the same amount of protein from each sample was separated by 12% SDS-PAGE. After blotting, the membranes were probed with anti-PTGER2 or anti-TIMP2 according to the manufacturer’s instructions. The experiments were repeated three times. GAPDH was used as the loading control. Analysis of the WBs was performed using ImageJ Software. (**C**,**D**) Optical density analyses of the target proteins after 24 h DFO treatment followed by 24 h FKN addition in a serum-free or serum-supplemented culture medium. (**E**,**F**) Optical density analyses of the target proteins after 48 h DFO treatment followed by 24 h FKN addition in a serum-free or serum-containing culture medium. The blots were cropped according to the molecular weight of the target protein. The original blots can be found in the Appendix A. The columns represent the mean ± SD of three independent experiments (*n* = 3). The asterisk shows *p* < 0.05 compared to the control. The cross means *p* < 0.05 compared to the DFO treatments. The double cross signs *p* < 0.05 compared to the F10 treatment. Abbreviations: DFO-desferrioxamine; FKN-fractalkine; F10-fractalkine 10 ng/mL, F20-fractalkine 20 ng/mL.

**Figure 7 ijms-24-07924-f007:**
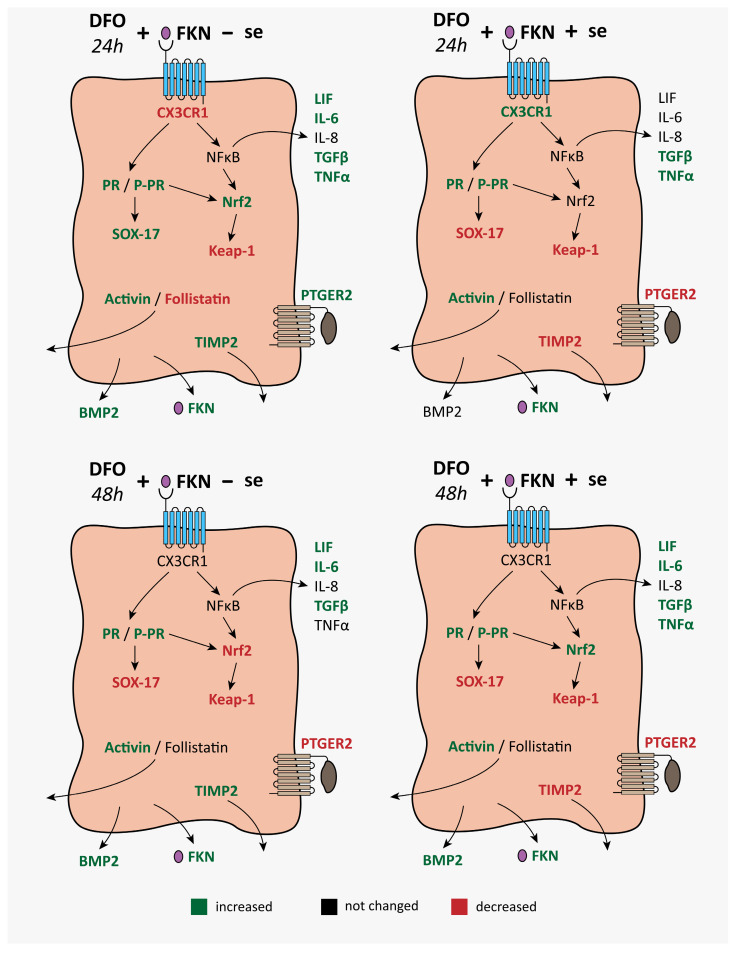
Schematic illustration of the alterations in the expression of the receptivity-related genes and proteins after 24 h or 48 h DFO treatments followed by 24 h long FKN supplementation in a serum-free culture medium or a serum-supplemented culture medium. CX3CR1 regulates the activity of PR, a regulator of the SOX-17 transcription factor. NFκB is one of the downstream signalling pathways of the FKN/CX3CR1 axis regulating the expression of the pro-inflammatory cytokines LIF, IL-6, IL-8 and TNFα. The FKN/CX3CR1 axis also controls the Nrf2-Keap-1 transcription factor system affecting the NFκB pathway and cytokine synthesis. DFO and FKN treatments also modify the expression of activin, follistatin, BMP2, FKN, PTGER2 and TIMP2.

**Table 1 ijms-24-07924-t001:** Real-time PCR primer list.

Primer	Sequence 5′→3′
activin A forward	GAACTTATGGAGCAGACCTC
activin A reverse	GGACTTTTAGGAAGAGCCAG
BMP2 forward	TAAGTTCTATCCCCACGGAG
BMP2 reverse	AGCATCTTGCATCTGTTCTC
CX3CR1 forward	CCATTAGTCTGGGCGTCTGG
CX3CR1 reverse	GTCACCCAGACACTCGTTGT
FKN forward	TACCTGTAGCTTTGCTCATC
FKN reverse	GTCTCGTCTCCAAGATGATT
follistatin forward	CAAAGCAAAGTCCTGTGAAG
follistatin reverse	CCTCTCCCAACCTTGAAATC
GAPDH forward	TGTTCCAATATGATTCCACCC
GAPDH reverse	CCACTTGATTTTGGAGGGAT
IL-1β forward	GAAATGATGGCTTATTACAGTGG
IL-1β reverse	GGTGGTCGGAGATTCGTA
IL-6 forward	CTGAGAAAGGAGACATGTAACAAGA
IL-6 reverse	GGCAAGTCTCCTCATTGAATC
IL-8 forward	CAGTGCATAAAGACATACTCC
IL-8 reverse	CACTCTCAATCACTCTCAGT
LIF forward	CTCGGGTAAGGATGTCTTC
LIF reverse	GGCGATGATCTGCTTATACT
Progesterone receptor A/B forward	CCAAAGGCCGCAAATTCT
Progesterone receptor A/B reverse	TGAGGTCAGAAAGGTCATCG
SOX-17 forward	CAGTATCTGCACTTCGTGTG
SOX-17 reverse	AGTAATATACCGCGGAGCTG
TGF-β forward	GCCAGAGTGGTTATCTTTTG
TGF-β reverse	GTAGTGAACCCGTTGATGT

## Data Availability

The data underlying this article are available in the article and its online Appendix A.

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
