# Peer review of "Fractalkine Improves the Expression of Endometrium Receptivity-Related Genes and Proteins at Desferrioxamine-Induced Iron Deficiency in HEC-1A Cells"

_ijms, 2023, doi:10.3390/ijms24097924_

Round 1

Reviewer 1 Report

This work proposes a seems that fractalkine ameliorates the negative effect of iron deficiency on the receptivity-related genes and proteins of HEC-1A endometrium cells, suggesting its important role in the regulation of endometrium receptivity.

This paper represent a further step in the research field of the authors, that have previously investigated the role of fractalkine on endometrium receptivity.

In my opinion, the quality of the manuscript is high in terms of characterizations performed, results and data interpretation, average in terms of readibility, process explanations and perspectives.

I would only recommend to the authors to include the term "anemia" not only in the conclusion, but also in the introduction and in the abstract, in order to give a better understanding also to a non-specialist audience.

Reviewer 2 Report

The current manuscript by Pandur et al explored the role of Fractalkine (FKN) using endometrial cells HEC-1A, emphasizing its role during iron deficiency.

Authors have published in related topics before. In 2020 authors published in this journal (IJMS) how FKN regulates expression of implantation-related genes using HEC-1A and JEG-3 trophoblast cells (PMID: 32365902, Int J Mol Sci. 2020 Apr 30; 21(9):3175). Recently, authors also published on how FKN influences endometrial receptivity and iron homeostasis using HEC-1A and JEG-3 co-culture system. Hence the rationale for the current study is justified.

Authors performed a fairly elaborate study, but some aspects should be addressed.

Main suggestions

1.      As authors performed study entirely using only one type of cells and only in vitro system, the obvious question is what happens in vivo. Authors should consider including an animal model to further verify the observations.

2.      A “limitations” section should be considered by the authors.

3.      Authors are suggesting that iron redistribution may play a role in the effect. Supplementary figure S1A shows the modulation of intracellular iron content by desferrioxamine (DFO) treatment. To confirm that serum bound iron is important in this context, authors are requested to repeat these measurements in presence and absence of serum.

4.      Authors are requested to provide more details on the FKN ELISA procedure, when the cells were also treated with the same protein (figures 2E and 2F). Are these values include the added proteins as well?

5.      Authors are requested to justify the choice of control for figure 1. Instead of comparing the expression to zero hour control, why not include cells without treatment of DFO at each time point?

Other comments

The inclusion of the full blots in the supplementary document is greatly appreciated. But please clarify how the molecular weights were assigned to the protein bands.

Reviewer 3 Report

The manuscript under review "Fractalkine improves the expression of endometrium receptivity-related genes and proteins at desferrioxamine-induced iron deficiency in HEC-1A cells" is devoted to the study of the protective properties of fractalkine in the effect of iron deficiency on HEC-1A cell culture, which is certainly relevant, given the high frequency of iron deficiency anemia in pregnancy and the negative role of proinflammatory cytokines, including intreleukin 6, in normal maternal fetal delivery. Overall, the study design is adequate, with new results on the protective role of fractalkin. The results concerning Nrf2 should have been supplemented with an analysis of its distribution between the cytosolic and nuclear fractions of cells. In the opinion of the reviewer, the discussion of the results should have been provided with a final scheme summarizing the obtained results.

Reviewer 4 Report

In this study, the authors investigated the effect of iron deficiency on the secretion of cytokines, chemokines and regulatory proteins in HEC-1A cells, which are involved in endometrial receptivity, and the potential modulating effect of fractalkine (CX3CL1/FKN) on these proteins. The study was conducted by treating HEC-1A cells with desferrioxamine (DFO) to induce iron deficiency, followed by the addition of FKN to the cells to evaluate its modulating effect.

Overall, the study suggests that FKN plays an important role in regulating endometrial receptivity and mitigating the negative effects of iron deficiency on the expression of receptivity-related genes and proteins in HEC-1A endometrial cells. The findings may have implications for understanding the mechanisms involved in embryo implantation and for developing new treatments for infertility related to endometrial receptivity.

However, it is important to note that the study was conducted in vitro using HEC-1A cells, which are a commonly used endometrial cancer cell line, and the results may not necessarily reflect the in vivo effects of FKN on endometrial receptivity in humans. Additionally, the study focused specifically on the effects of iron deficiency and FKN on certain cytokines, chemokines, and regulatory proteins related to endometrial receptivity, and it is unclear how these findings may relate to other factors involved in embryo implantation and fertility. Further research is needed to confirm and expand upon these findings in a more physiological setting.

Reviewer 5 Report

For the most part, the manuscript is carefully and thoroughly prepared. There are some shortcomings that the authors need to correct and improve. In the introduction, the authors should quote publications from the last five years showing the fact that the conducted research concerns problems that are important for the development of science. I conclude they were prepared extremely cursorily. Bearing in mind all the experiences carried out, the conclusions should be developed and expanded. 

The language of the publication should be checked by professionals.

Round 2

Reviewer 4 Report

 Accept in present form.